# A Focal Impact Model of Traumatic Brain Injury in *Xenopus* Tadpoles Reveals Behavioral Alterations, Neuroinflammation, and an Astroglial Response

**DOI:** 10.3390/ijms23147578

**Published:** 2022-07-08

**Authors:** Sydnee L. Spruiell Eldridge, Jonathan F. K. Teetsel, Ray A. Torres, Christina H. Ulrich, Vrutant V. Shah, Devanshi Singh, Melissa J. Zamora, Steven Zamora, Amy K. Sater

**Affiliations:** 1Department of Biology and Biochemistry, University of Houston, Houston, TX 77204, USA; slspruie@central.uh.edu (S.L.S.E.); jfteetse@cougarnet.uh.edu (J.F.K.T.); r.torres226@yahoo.com (R.A.T.); christina_ulrich@takarabio.com (C.H.U.); vvshah@mdanderson.org (V.V.S.); devanshisingh397@yahoo.com (D.S.); melissa.jo.zamora@gmail.com (M.J.Z.); smzamora87@gmail.com (S.Z.); 2Takara Bio, 2560 Orchard Parkway, San Jose, CA 95131, USA; 3Department of Genetics, M.D. Anderson Cancer Center, Houston, TX 77030, USA; 4Texas College of Osteopathic Medicine, University of North Texas Health Science Center, Forth Worth, TX 76107, USA; 5Memorial Family Residency Program, 14023 Southwest FWY, Sugarland, TX 77478, USA; 6Baker Hughes, Houston, TX 77073, USA

**Keywords:** TBI, brain injury, inflammation, astrocyte, microglia, *Xenopus*, tadpole

## Abstract

Traumatic Brain Injury (TBI) is a global driver of disability, and we currently lack effective therapies to promote neural repair and recovery. TBI is characterized by an initial insult, followed by a secondary injury cascade, including inflammation, excitotoxicity, and glial cellular response. This cascade incorporates molecular mechanisms that represent potential targets of therapeutic intervention. In this study, we investigate the response to focal impact injury to the optic tectum of *Xenopus laevis* tadpoles. This injury disrupts the blood-brain barrier, causing edema, and produces deficits in visually-driven behaviors which are resolved within one week. Within 3 h, injured brains show a dramatic transcriptional activation of inflammatory cytokines, upregulation of genes associated with inflammation, and recruitment of microglia to the injury site and surrounding tissue. Shortly afterward, astrocytes undergo morphological alterations and accumulate near the injury site, and these changes persist for at least 48 h following injury. Genes associated with astrocyte reactivity and neuroprotective functions also show elevated levels of expression following injury. Since our results demonstrate that the response to focal impact injury in *Xenopus* resembles the cellular alterations observed in rodents and other mammalian models, the *Xenopus* tadpole offers a new, scalable vertebrate model for TBI.

## 1. Introduction

Traumatic brain injury (TBI) is the primary cause of disability and is now considered the third leading cause of death globally [1]. In the U.S., between 3 and 5 million people are affected by TBI each year, and the long-term impacts often result in significant functional limitation, including cognitive decline, memory loss, and difficulty with attention, behavior, and emotional regulation [2]. Despite the prevalence of TBI, there are remarkably few treatments that are effective in promoting recovery or mitigating neurological damage. Moreover, emerging evidence suggests that repeated mild TBI may result in significant impairment over time; even a single injury can lead to both acute and chronic symptoms. To understand and address the consequences of TBI, it is essential to consider TBI as the initiation of a disease process, rather than a single, isolated event.

Experimental studies have delineated key features of the cellular and physiological damage resulting from TBI, revealing distinct acute and chronic phases of the response to injury. The acute, or primary, injury results from the direct mechanical insult or rapid acceleration, producing axonal shearing, neurotoxicity, and apoptosis. The extent of this primary injury can be either focal or diffuse. This primary injury generates a secondary injury cascade, arising as a complex series of interacting molecular pathways, characterized by excitotoxicity, oxidative stress, neuro-inflammation, and BBB disruption. Within minutes of injury, the release of proinflammatory cytokines (IL-1β, TNFα, IL-6) and other mediators of inflammation establishes a neuroinflammatory microenvironment that stimulates the activation of resident glial cells, as well as the infiltration of non-native immune cells into the injured brain [3].

This neuroinflammatory response incorporates positive feedback between microglia, the resident macrophages of the brain, and astrocytes, which undergo a significant phenotypic alteration known as the reactive astrocyte response. Astrocyte reactivity incorporates cell migration into the injury site, morphological alterations, upregulation of glial fibrillary acidic protein (GFAP) and other intermediate filament (IF) proteins, release of inflammatory cytokines, and re-entry into the cell cycle reviewed in [4,5]. These changes culminate in reactive astrogliosis, which limits the spatial extent of injury but can also restrict neuroregenerative responses. The persistence of neuro-inflammation prolongs the secondary phase of injury, hindering recovery from TBI.

Mammalian models have been essential in revealing the complex and often heterogeneous pathophysiology of TBI. The cellular consequences and behavioral sequelae of brain injury have been investigated primarily through the use of controlled cortical impact (CCI), fluid percussion (FP), or weight drop-impact injury models in rat, mouse, and pig reviewed in [6,7]. In addition, in vitro models of cellular response to injury have identified many candidates for therapeutic intervention, although few of these have proved effective in in vivo studies; at present, there are no pharmacological therapeutics for TBI [6,7]. Moreover, prevailing mammalian animal models are often not feasible for large-scale pharmacological studies. There is thus a pressing need for animal models of TBI that can be scaled for high-throughput screens to identify pharmacological compounds that can elicit tissue repair or mitigate the impacts of secondary injury following TBI.

The *Xenopus* tadpole offers a range of strengths that complement the advantages of mammalian systems used in studies of TBI. The developmental organization of the central nervous system is conserved across vertebrates, and *Xenopus* tadpoles have emerged as a powerful model for neurodevelopmental disorders [8,9,10,11]. A single round of fertilization can yield several hundred tadpoles, and *Xenopus* has proved effective for chemical or pharmacological screens [12]. Most human genes have *Xenopus* orthologues, and the *Xenopus* genome bears remarkable synteny to the human and mouse genomes. Additionally, genome editing via CRISPR/Cas-based strategies is straightforward and scalable. Finally, because *Xenopus* tadpoles at NF stage 55 lack a dorsal skull, there is no need for a craniotomy prior to the administration of a focal impact injury; a craniotomy is a necessary initial step in many mammalian models of TBI and is objectively damaging on its own [13].

We have developed a focal impact injury model as a foundation for studies of TBI in *Xenopus laevis* tadpoles. Tadpoles are injured at the optic tectum, the homologue of the mammalian superior colliculus, which acts as the visuomotor processing center of the *Xenopus* brain [14]. We have assessed the response to injury in terms of behavior, including visual responsiveness, while evaluating the extent of pathophysiological damage and inflammatory response. We find that focal impact TBI in *Xenopus laevis* exhibits pathophysiological and behavioral responses that are comparable to mammalian TBI and thus offers a promising and scalable model system for investigation of TBI.

## 2. Results

### 2.1. Focal Impact Injury Model

We developed a focal impact injury system that delivers a reproducible impact of 2.5 lbf across an area of 0.78 mm^2^ (corresponding to 208 psi) to the dorsolateral midbrain of a *Xenopus laevis* tadpole. Briefly, tadpoles are anesthetized in MS-222 and transferred with a plastic scoop to a cradle under a dissecting microscope. The impact is delivered via a pneumatic device controlled by a foot pedal; the piston has a 3-D printed tip so that the size of the contact area can be altered. Once the injury has been administered, the tadpole is transferred to a recovery tank. The total time out of water is typically less than 90 s. Sham-treated tadpoles are anesthetized, transferred to the cradle, and then moved to a recovery tank. Tadpoles are monitored at frequent intervals during the recovery from anesthesia, 20–30 min. Recovery is marked by a restoration of responsiveness to touch, a return of tail movement, and normal buccal pumping.

### 2.2. Initial Studies and Validation

We carried out an initial set of studies on younger tadpoles (st. 48); at this stage, tadpoles show a significant capacity for regeneration [15], and the brain is considerably smaller than at NF st. 55 (Figure A1). We began with work at this stage so that we could compare our findings with those of earlier studies of neural regeneration [15]. These findings are summarized in Appendix A Figure A1. Focal impact injury resulted in axonal damage (Figure A1A,B) and a general reduction in activity in open-field tests (Figure A1D,E). Quantitative RT-PCR assays showed elevated expression of *timp1* and *steap4*, two genes known to be upregulated in reactive astrocytes in mice (Figure A1C) [16]. To validate the increase in reactive astrocyte genes, we used an alternative injury model involving transient exposure to elevated pressure (up to 0.5–1 psi for <5 min). Expression of reactive astrocyte genes was moderately elevated (Figure A1F), and open field activity scores were decreased in pressure-treated tadpoles (Figure A1G,H).

Thus, both focal impact injury and pressure injury lead to decreased activity and elicit an increase in two genes involved in the reactive astrocyte response in pre-metamorphic tadpoles. Once these foundational studies were complete, we sought a closer correspondence with mammalian brain injury and thus carried out subsequent injury studies at st. 55, when the brain is considerably larger (Figure A1B) and the capacity for regeneration in the brain is greatly reduced. We discontinued the use of the pressure model when we switched to older tadpoles, as the pressure system did not easily accommodate the increased size of st. 55 tadpoles.

### 2.3. Survival

We assessed survival following focal impact injury in st. 55 tadpoles. Under these conditions, we found that over 98% of tadpoles survive both injury and anesthesia. We saw little difference between survival of injured vs. sham-treated tadpoles (Table 1). Tadpoles are highly sensitive to anesthesia, and overlong exposure to MS-222 at this concentration can be fatal. Injured tadpoles typically survive up to at least 7 days; only 2% of injured tadpoles die within 72 h of injury.

### 2.4. Disruption of the Blood-Brain Barrier

We carried out intraventricular (i.v.) injections of sodium fluorescein to evaluate the integrity of the blood-brain barrier (BBB) after injury. Following injury, tadpoles were kept for varying intervals before anesthesia and injection. After tadpoles were anesthetized, they were injected in the fourth ventricle of the brain with approximately 10 nL 0.1 mg/mL sodium fluorescein (NaF) fluorescent tracer, as described by De Jesus Andino and colleagues [17]. Diffusion of fluorescein was visualized by fluorescence microscopy immediately following the i.v. injection as an indicator of blood-brain barrier (BBB) dysfunction.

In the sham-treated tadpoles, NaF is retained within the BBB at all time points (Figure 1A,C). In contrast, injured animals show significant diffusion of NaF across the BBB into surrounding tissue (Figure 1B,D). All injured tadpoles exhibited evidence of BBB leakage at 3 h, 24 h, and 48 h post-injury (h.p.i), whereas the fluorescein was restricted to the brain in the sham tadpoles. These results demonstrate that focal impact injury compromises the integrity of the BBB, a known pathological consequence of TBI.

### 2.5. Edema

We next evaluated the extent of brain swelling, or edema, following focal impact injury. At specific intervals following injury, tadpoles were euthanized in order to isolate and weigh the brains. The isolated brains were then dried under vacuum and reweighed to obtain the dry weight. The difference between the wet and dry weights reveals the water content of the brains, as discussed in the Methods section. These experiments revealed that the water content of the brain increases following injury, allowing the extent of edema to be quantified (Figure 1E). The degree of brain edema is highest at 48 h following injury, peaking at an increase of ~2%. It then begins to resolve, and the water content of the injured brain is statistically indistinguishable from the sham-treated brain by 72 h post-injury. The extent of injury-induced edema was very similar to increase in brain water content observed in adult male Sprague-Dawley rats subjected to controlled cortical impact injury [18] and in male mice subjected to fluid percussion injury [19].

### 2.6. Behavior

We assessed the behavioral sequelae of focal impact injury using several established assays (Figure 2); in each case, we compared injured tadpoles with sham-treated controls. For each behavioral study, our findings represent at least 18 individuals across at least two clutches of tadpoles. First, we used open field assays to assess the level of activity and alterations to normal swimming behavior, recording individual tadpole behavior over a 10-min period at timepoints of 0, 24, 48, and 72 h post-injury; tadpoles are recorded prior to anesthesia and injury, and this recording serves as the “0” baseline timepoint. From subsequent recordings, we quantified the number of rapid reversals of direction, referred to as C-starts [20], within the observation period. Tadpoles subjected to focal impact injury showed significantly fewer C-starts than sham controls 24 hrs after injury, and this difference persisted up to 72 h post-injury (Figure 2A).

Since the focal impact injury targets the optic tectum at the dorsolateral midbrain, we sought to determine how this injury affected behavior in response to visual stimuli. We used two assays to examine the effects of injury on visually responsive behaviors. First, we used a Visual Preference Assay, which is based on the finding that *Xenopus* tadpoles prefer dark environments over light ones [21]. Tadpoles are introduced on the light side of a 12-cm round chamber that consists of light and dark halves (“light/dark box”). They are then recorded over a 10-min interval, which allows for quantification of the amount of time spent in the light vs. dark halves. Sham-treated tadpoles spend nearly 1/3 of the recorded interval on the light side, as do “T0” tadpoles prior to anesthesia and/or injury (Figure 2B). In contrast, after 24 h, injured tadpoles spend significantly greater time on the light side than do sham tadpoles, and this difference persists over 72 h, similar to the difference observed in C- starts.

We also used the Spot Avoidance Assay developed by Cline and colleagues [15] to investigate visually responsive swimming behavior. This assay incorporates the observation that tadpoles will change their swimming trajectory to avoid a moving spot of light. Tadpoles are placed in tanks on a transparent plexiglass platform situated above a projector that presents them with a dynamic pattern of light spots on a dark background; we record swimming behavior until each animal has experienced 10 interactions with the visual stimulus. We then evaluate the frequency of avoidance behavior during the first 10 encounters with a moving spot to establish an “avoidance index”, that represents the fraction of times that an avoidance response was performed out of the 10 recorded interactions with the stimulus.

We compared spot avoidance between injured and sham-treated tadpoles at daily intervals over a 7-day period (Figure 2C). At 24 h.p.i., the avoidance index for injured tadpoles is less than half that of sham-treated tadpoles. The avoidance index for injured tadpoles increases gradually over subsequent days, until by Day 6, when it has recovered to 80% of the level of the sham controls. Taken together, these findings indicate that focal impact injury impairs general behavior, and that it specifically disrupts the visuomotor processing response to visual stimuli, which is directly mediated by the optic tectum.

### 2.7. Tissue Damage and Cellular Response to Injury

The tissue-level consequences of injury can be visualized via immunofluorescence and confocal imaging. In sham-treated brains, DAPI staining reveals a tight laminar organization above the ventricular surface, particularly around the lateral extensions of the ventricle. Few nuclei are visible within the neuropil layer, and the outer surface includes a layer of vimentin-positive cells that we identify as radial glia. In contrast, midbrains subjected to focal impact injury show abrasion and damage at the outer surface, loss of axonal material (Appendix A, Figure 1A), and expansion and disruption of the multicellular “laminae” surrounding the lateral aspects of the ventricle by 3 h following injury (Figure 3A vs. Figure 3D; Figure 4C,E). In injured brains, nuclei accumulate within the neuropil in the marginal layer at and around the site of injury (Figure 4E and Figure A1).

Both astrocytes and microglia play critical roles in the response to neural injury. We investigated the distribution of these cells in the tadpole midbrain following focal impact injury. Astroglial cells were visualized by immunofluorescent detection of the astrocyte protein Aldh1L1 and the astroglial protein vimentin (Figure 3 and Figure 5A–F). These studies were carried out in tadpoles carrying an N-tubulin-GFP transgene [22], so the neurons are also visible. In sham-treated brains 3 h following treatment, astrocytes are relatively sparse, appearing primarily near the ventricles (Figure 3A). Astrocytes in sham-treated midbrains show limited ramification, often showing a distinctive “triskelion” shape (Figure 3B,H); these cells can be visualized with antibodies directed against Aldh1L1. While astrocytes are rare in the dorsolateral midbrain, radial glia are visible both at the ventricular layer and throughout the dorsolateral neuropil layer.

Injured midbrains display a very different distribution of these cells, beginning at 3 h post-injury. Astrocytes are distributed throughout the dorsolateral midbrain, primarily within and around the injured area, but also in the contralateral optic tectum. These cells also show a change in morphology: injured midbrains contain astrocytes that are either bipolar or amoeboid in shape (Figure 3E,K). The enlarged amoeboid astrocytes are often clustered close to the injury site, while the bipolar cells are located at a greater distance from the injury site, or on the contralateral side. Some clustered amoeboid astrocytes are co-labeled with N-tubulin-GFP (Figure 5D–F), indicating the possibility that astrocytes are participating in the phagocytosis of neurons. At 24 h following injury, astrocytes are primarily round, rather than ramified. The anti-vimentin immunofluorescence shows a denser and more closely aligned distribution in the injured tissue, presumably reflecting alterations in astroglial activity.

We evaluated microglial dynamics after fluorescent labeling of microglia via intraventricular injection of IB4-Alexa 647 (Figure 4 and Figure 5G–J). Very few microglia are observed in sham midbrains, and they are seen only close to the ventricular layer. The optic tectum itself is devoid of microglia. By 3 h after injury, however, microglia are distributed in the injured area and surrounding regions, and few microglia remain near the ventricles (Figure 4C–E). As with astrocytes, a few microglia are co-labeled with IB4 and N-tubulin-GFP, suggesting that these microglia have phagocytosed axonal material (Figure 5G–J). Such co-labeled cells were rarely observed in sections of sham-treated midbrains.

At 24 h following injury, microglia are distributed closer to the ventricular layer of the midbrain (Figure 4I–K). A small number of microglia are co-labeled with either N-tubulin-GFP and/or vimentin, suggesting phagocytosis of axons or radial glia (Figure 5G–J). Microglia form linear aggregates extending from the ventricular layer out toward the neuropil.

### 2.8. Astroglial Response to Injury

Since our preliminary studies (Figure A1) suggested that *Xenopus* tadpoles undergo a reactive astrocyte response, we investigated alterations in the expression of several astroglial genes during the response to focal impact injury. Injured and sham-treated tadpoles were allowed to recover for different intervals and then euthanized; the midbrains were dissected and processed for RNA isolation, cDNA synthesis, and quantitative RT-PCR.

We first evaluated a set of genes associated with astroglial function: *fabp7, eeat1* (referred to as *glast*), and *vimentin*, as well as *aldh1l1,* associated with mature astrocytes, *nestin*, expressed in radial glia (Figure 6F–K), and *aquaporin-4*, an astroglial water channel that has been implicated in the formation of edema. Some of these genes showed increased expression by the 24 h post-injury (*aldh1l1, GLAST, vimentin, aquaporin-4*), returning to baseline within 7 days. Others showed a more delayed increase in expression (*nestin, fabp7)*.

These effects on astroglial genes provide a frame of reference for analysis of genes associated with the reactive astrocyte response (*timp1, steap4*, Figure 7A,B) or neuroprotective activity: *bdnf* (brain-derived neurotrophic factor; [23]), *manf* (Mesencephalic astrocyte-derived neurotrophic factor [24], *clusterin* (e.g., [25]), and ubiquitin carboxy-terminal hydrolase L1 *(uchl-1)* [26] (Figure 7C–F). Although *timp1* expression was moderately elevated after 24 hrs, *steap4* expression remained stable throughout the post-injury interval. These findings contrast with our preliminary observations of an increase in expression in both genes in whole brains following focal impact injury, suggesting that the reactive astrocyte response in different regions of the brain may be mediated by distinct transcriptional responses. There are signs of response to injury in regions at some distance from the injury site, presumably reflecting the propagation of force through the brain tissue. In the telencephalon, microglia accumulated near the periphery on the ipsilateral, but not the contralateral, side (Figure 8A,B), persisting at 48 h post-injury.

The neuroprotective genes showed a similar heterogeneity. While expression of *manf* was moderately elevated by 3 h.p.i., expression of *bdnf* was increased only at 48 h. Levels of *clusterin* remained near baseline through 72 h, and were elevated at 168 h.p.i.

### 2.9. Inflammation

We used a similar approach to evaluate the inflammatory response over time. Quantitative RT-PCR assays show that the inflammatory cytokines *TNFa*, *IL-1b*, and *IL-6* are dramatically upregulated within 3 h of injury (Figure 9A–C). Expression of these cytokines declines but remains moderately elevated at 24 h; by 48 h, cytokine expression is at near-baseline levels. Focal impact injury also elicits increased expression of *NF-kB* and *mmp9*, which has decreased by 48 h (Figure 9D,E). Arginase (arg) is essential for the regulation of nitric oxide (NO) [27], a component of injury-associated neuroinflammation, and *arg* expression is elevated through 48 h (Figure 9F).

### 2.10. Microglial Gene Expression

Since microglia infiltrate the injured tissue by 3 h after injury, we expected that microglia-associated gene expression would be increased by this time point. Expression of *csf1R* paralleled the appearance of microglia at the injury site (Figure 8B,C). In contrast, two other microglia-specific markers, the ADP receptor *P2RY12* (Figure 8D) and *tmem119* (Figure 8E), show distinct temporal profiles. Although *P2RY12* expression peaks sharply at 24 h, returning to baseline by 48 h, *tmem119* shows modest and variable changes in expression returning to baseline within 72 h. These results do not correspond to the elevated numbers of microglia observed in the injured brains, suggesting that expression levels may reflect transcriptional responses to the cellular environment.

## 3. Discussion

We have developed a new vertebrate model for traumatic brain injury, suitable for cellular, molecular, and behavioral studies. Our findings demonstrate critical parallels with established rodent models, including cell and tissue-level changes such as edema [28], disruption of the blood-brain barrier [29], inflammation [30,31], and a limited reactive astroglial response [32]. We also observe reversible alterations in behavior, both in visually-directed behavioral responses and overall activity.

We focused on visually-directed behavior because injury to the optic tectum is expected to disrupt the processing of visual inputs and visually-driven motor responses [14]. This approach was pioneered in earlier studies from Cline and colleagues, which investigated neural regeneration in the optic tectum of pre-metamorphic tadpoles using a surgical ablation injury model [15,33]. These studies showed that local injury in the dorsolateral midbrain leads to a sharp decline in visual avoidance. The capacity for visual avoidance recovers gradually and has returned to baseline at 7 days post-injury. Recovery of distinct behavioral responses requires proliferation of neural progenitors to replenish neurons at the injury site [15], as well as NMDAR-dependent neuronal activity to integrate new neurons into existing neural circuits [33]. While this work elucidates the neural regenerative response characteristic of pre-metamorphic tadpoles, regenerative capacity is essentially lost with the activation of thyroid hormone (TH)-dependent processes at metamorphosis (reviewed in [34]. The time course of behavioral changes in response to injury observed following focal impact injury parallels the temporal profile of behavioral recovery in pre-metamorphic tadpoles subjected to stab injury. The similarity in the kinetics of the behavioral response despite the difference in regenerative capacity suggests that persistent neuroprotective mechanisms may contribute to neural regeneration and recovery.

Our findings reveal an astroglial response to injury: astrocytes in injured brains show a rounded amoeboid morphology and appear to migrate into the site of injury. Confocal imaging also reveals increased accumulation of *aldh1l1* and *vimentin* throughout injured brain tissue, and expression of *vimentin* and other astroglial genes is moderately elevated. In addition, numerous Aldh1l1^+^ cells show punctate co-labeling for N-Tubulin-GFP, suggesting that these are phagocytic astrocytes removing damaged axonal material, similar to the phagocytic astrocytes observed following injury to the *Xenopus* tadpole optic nerve [35]. Although two genes associated with the reactive astrocyte response, *timp1* and *steap4*, are upregulated in whole brains from slightly younger tadpoles subjected to focal impact injury, increased expression of these genes in injured midbrains is variable. *Timp1* and *steap4* were initially selected from the set of genes identified by the Barres lab [16] as being upregulated in reactive astrocytes because they are strongly conserved in *Xenopus*. Many of the genes in this set function in inflammation or innate immunity, and identification of corresponding *Xenopus* orthologues for such rapidly evolving genes was difficult. Reactive astrogliosis is a central feature of the mammalian response to neural injury [4], and zebrafish also undergo limited astrogliosis [36]. Transcriptomic studies currently in progress should allow us to delineate the extent of reactive astrogliosis in *Xenopus*, in accordance with the framework and caveats presented in [37].

Our findings also show a rapid and sustained accumulation of microglia within the injured tissues: microglia appear to form clusters that extend between the ventricular regions and the periphery, and microglia appear on the ipsilateral side of the telencephalon within 48 h after injury, presumably as a result of mechanical impact spreading to more distant areas throughout the brain. We see punctate accumulation of N-tubulin-GFP within microglia, implicating the microglia in phagocytic removal of injured axons, as with similarly labeled astrocytes. This observation is consistent with the scavenging functions of mammalian microglia in neural injury [38]. *Xenopus* midbrain microglia have also been shown to engage in trogocytosis (i.e., partial phagocytosis) during axon pruning in vivo [39].

Secondary injury in TBI is a complex process involving damage-associated molecular patterns (DAMPs), cytokines and chemokines, neurons, microglia, and astrocytes, all of which are involved in neuroinflammation [40,41,42]. Neuroinflammation governs many aspects of recovery and repair following neural injury [43], and the neuroinflammatory response emerges primarily from the coordinated activity of reactive astrocytes and microglia [44,45]. Inflammation is also known to accelerate the breakdown of the BBB, increase edema, and promote reactive astrogliosis [46,47]. We propose a model (Figure 10) in which the acute phase of TBI is accompanied by an early peak in pro-inflammatory mediators and cytokines, followed by a rapid microglial response; these early responses act in concert to promote reactive astrogliosis, which peaks within the same interval as edema. TBI induces the early production and release of cytokines and chemokines by a diverse array of immunocompetent cell types such as microglia and astrocytes [48,49,50]. Our current findings suggest a similarly timed upregulation of pro-inflammatory cytokines (TNF-α, IL-1β, IL-6) at 3 h.p.i., along with elevated expression of *nf-kB* and other mediators of the inflammatory response. Both microglia and astrocytes participate in amplifying the neuroinflammation response, and prolonged activation of either cell type may establish a positive feedback loop, sustaining the secondary injury cascade and promoting additional neuronal cell loss [38,41,45,51]. While primary injuries are often difficult to prevent and result in rapid cell death, the secondary injury cascade develops in a progressive manner; thus, secondary injury represents a promising target for therapeutic intervention [52].

Edema arises from the secondary injury cascade initiated following TBI; it generally occurs within hours after the initial injury and has recently been identified as an important prognostic indicator of mortality and disability in TBI patients [53,54]. In the current study, we found that a focal impact injury to the optic tectum produces disruption of the blood-brain barrier (BBB), edema, and upregulation of *aqp4*, which encodes one of the most abundant water channel proteins in the brain. AQP4 channels are highly expressed in astrocyte endfoot processes and are critical for the regulation of water movement to and from the brain parenchyma [55]. Increased expression of *aqp4* is a known consequence of TBI [56,57] and is positively correlated with increased edema and worsened neurological deficits [58,59]. Our findings indicate that edema peaks at 48 h.p.i. (Figure 1E and Figure 10), which corresponds with the peak in *aqp4* gene expression (Figure 6H). We also observed disruption of the blood-brain barrier at both 24 and 48 h.p.i. (Figure 1A–D). AQP4 is involved in cytotoxic edema, as well as vasogenic edema [59], which results from disruption of the blood-brain barrier [60], and AQP4 knockout mice display exacerbated levels of vasogenic edema [61,62]. In view of these observations, we speculate that the elevated expression of *aqp4* at 48 h may facilitate the subsequent reduction in edema.

At 72 h.p.i., when *aqp4* gene expression levels are still elevated, edema has begun to decline. Our findings support a potential relationship between increased *aqp4* gene expression, edema, and BBB dysregulation.

The use of the *Xenopus* tadpole as a model for TBI offers several advantages that complement the considerable strengths of the commonly used rodent models. In particular, this system is scalable and is thus suitable for large-scale discovery screens to identify pharmacological compounds that could either limit neurotoxicity, modulate reactive astrogliosis, or promote neuroprotective responses. Such screens could potentially reveal new avenues for therapeutic development or identify beneficial effects for compounds approved for other treatments.

While zebrafish and other teleost models offer similar opportunities for scalability [63], teleosts retain the capacity for neural regeneration in adulthood [64]. In contrast, the loss of regenerative capacity in the late pre-metamorphic *Xenopus* tadpole provides a closer resemblance between the tadpole brain and the mammalian brain; in humans, recovery from injury must depend on neuroprotective or immunomodulatory processes, and the *Xenopus* tadpole brain reflects comparable limitations. Both *Xenopus* and zebrafish provide scalable and cost-effective systems that can be used to expand the scope of pharmacological investigation and treatment for traumatic brain injury.

## 4. Materials and Methods

### 4.1. Tadpole Husbandry

*Xenopus laevis* embryos were reared to Nieuwkoop and Faber [65] stage 55 in treated tank water with aeration, under husbandry conditions as described in [66]. Tadpoles were fed daily with a slurry of Sera-Micron.

### 4.2. Focal Impact Injury

*Xenopus laevis* tadpoles at St. 55 are subjected to either sham treatment or focal impact injury via the following steps. Sham-treated tadpoles are anesthetized for 75 s in buffered 0.05% tricaine methanosulfate (MS-222, Sigma Aldrich A5040, St. Louis MO, USA) buffered to pH 7.4, transferred to a dish of tank water for approximately 40 s, then moved into the injury cradle, held briefly, and then returned to tank water to recover from anesthesia. Injured animals undergo anesthesia and serial transfer through tank water into the injury cradle. Once they are placed and oriented in the injury cradle, they are subjected to focal impact injury to the dorsolateral midbrain on either the left or right side. The injury is delivered via a pneumatic piston device with a 3-D printed tip; the device is connected to a CO_2_ tank and released with a foot pedal to administer an impact of 2.5 lbf across a tip diameter of 0.78 mm^2^. More detailed methods are provided in Appendix B. Once injured, the tadpole is returned to a tank for monitoring and recovery. The process of administering the focal impact injury takes approximately 2 min.

### 4.3. Intraventricular Injection

To assess the permeability of the Blood-Brain Barrier, tadpoles were anesthetized with 0.05 g/L MS-222 for 75 s prior to injection. Tadpoles were injected with sodium fluorescein (1 µg/mL; Sigma Aldrich F6377) in the 4th ventricle with until fluorescein could be seen throughout the ventricles. To visualize microglia, tadpoles were injected with Isolectin GS-IB4 647 (ThermoFisher I32450, Waltham, MA, USA) 48 h prior to observation. The GS-IB4 647 was diluted in 0.1% DMSO to a final concentration of 500 µM. Tadpoles were anesthetized as described previously and then injected with approximately 65 nL into the 4th ventricle of an anesthetized tadpole.

### 4.4. Immunofluorescence

For the immunofluorescence studies, tadpole heads were fixed overnight in 4% PFA. The brains were then dissected, and transferred through a series of overnight incubations in 10%, 20%, and 30% sucrose at 4 °C. Tadpole brains were embedded in OCT (Tissue-Tek, 4583, Sakura Finetek, Torrance CA, USA) in Tissue-Tek Cryomolds positioned for transverse sectioning, flash frozen in precooled 75% ethanol, and stored at −80 °C. Samples were sectioned using a Leica CM 1950 Cryostat at 12–20 µm. Sectioned samples were first heated at 37 °C for 30 min on a slide warmer following sectioning, then stored at −20 °C until staining.

Sections were permeabilized in phosphate-buffered saline (PBS; pH 7.4) containing 0.2% Triton X-100 (PBST), then incubated in blocking solution (TBS, 2 mg/mL BSA, and 20% normal goat serum) for 1.5 hrs at room temperature. TBS + 2 mg/mL BSA (BSA, Sigma, cat #A7906) was used as the buffer for all further incubations and washes. Sections were incubated in primary antibody solution, diluted in TBS + 2 mg/mL BSA overnight at 4 °C on a nutator. The following primary antibodies were used: Mouse α-Aldh1L1 (1:200, ab56777-Abcam, Cambridge UK), Rabbit α-Vimentin (1:200, ab16700-Abcam), and Chicken α-GFP (1:2000, ab13970-Abcam). Samples were then incubated for 1–2 h at room temperature with secondary antibodies, diluted in 1X TBS + 2 mg/mL BSA. Secondary antibody dilutions were made using: Goat α Mouse, Alexa 647 (1:200, ab150127-Abcam), Goat α Rabbit, Alexa 594 (1:200, ab150115-Abcam), Goat α Chicken, Alexa 488 (1:200, ab150169-Abcam), and DAPI (1:1000, #62248-Thermo Scientific). After staining, sectioned samples were covered using Vectashield Mounting Medium for fluorescence (product # H1000) and imaged on an Olympus FV3000 inverted confocal microscope in the University of Houston Biology and Biochemistry Imaging Core.

### 4.5. Evaluation of Brain Edema

Brain edema was quantified using established methods of wet/dry weight collection [67,68]. After MS-222 euthanasia, brains were dissected from both injured and sham groups at timepoints of 3, 24, 48, and 72 h.p.i. (*n* = 15/timepoint). Immediately following removal, the brains were placed in a pre-weighed container, weighed, and then dried overnight in a vacufuge at 30 °C and reweighed. The weight before dehydration is labeled the “wet weight” and the weight after dehydration is labeled the “dry weight”. Brain water content is then calculated using the following equation:Water content (%) = ((wet weight − dry weight)/wet weight) × 100

### 4.6. RNA Isolation and Quantitative RT-PCR

Isolated midbrains were lysed in TRIzol (ThermoFisher 15596026). RNA was then extracted using the Zymo Direct-zol RNA MiniPrep system (Zymo R2050, Irvine CA, USA) according to manufacturer’s instructions. RNA was reverse transcribed using SuperScript III (ThermoFisher 12574026) according to manufacturer’s instructions. Quantitative Reverse-Transcriptase-Polymerase Chain Reaction (Q-RT-PCR) assays were carried out as described in [69] and analyzed using the ΔΔcT method [70]. Primers for Q-RT-PCR assays are provided in Table A1; gene selection and primer design were based on genomic resources provided by Xenbase [71]. Q-RT-PCR reactions were carried out using 200 ng cDNA per reaction. Relative values for genes of interest were identified by normalization to the geometric mean of values for the housekeeping genes Histone H4 (*HisH4*) and ornithine decarboxylase (*ODC*) to obtain the ΔcT values, followed by comparison of ΔcT values between control and experimental samples to obtain the ΔΔcT values. Fold change for each target gene was then calculated from ΔΔcT.

### 4.7. Behavioral Assays

Behavioral assays began at 0 h post-injury (h.p.i.), which represents baseline data collected prior to the administration of anesthesia or injury. All tests were then carried out on individual tadpoles for three to seven consecutive days following treatment, at the same time each day. All behavioral assays were recorded and analyzed using Noldus EthoVision Tracking Software (Version #13) and camera system. R Studio was used for all statistical analysis. Where possible, data analysis was conducted blind to treatment. All behavioral assays were analyzed using two-way ANOVAs, followed by Tukey’s post hoc test when appropriate. Power analyses were performed in R Studio to determine the sample size. Statistical significance was set at *p*  ≤  0.05.

*4.7a—Open Field:* Tadpoles were tested in the open arena field to assess locomotor and visual behaviors coordinated by the optic tectum [72]. Each tadpole was individually placed in a circular, open arena 12 cm in diameter, and allowed to explore freely for 10 min. *C-Starts*, an innately performed escape swimming behavior [20], are quantified using EthoVision. The *C-Start* index is calculated for each individual timepoint (*CT*) using each individual’s baseline (0 hpi, *C*0) data:
$$C-Start\;Index=\frac{\left(CT-C0\right)}{C0}\;$$

*4.7b—Visual Preference (Light/Dark Box):* As a common behavioral test [73], this assay has been previously modified for other aquatic organisms such as zebrafish [74]. In this study, light/dark box assays were performed to assess the retention or loss of visual preference in response to injury, using a 12-cm circular arena containing frog water to a depth of 1 cm. Half of the arena was brightly lit by a fluorescent white light, and this half has a white bottom and sides. The opposite side of the arena was covered to protect from direct light and has a black bottom and sides. Tadpole swimming behavior was recorded for 10 min, time spent in the light chamber was quantified using EthoVision, and a percentage score was calculated for the time spent in the light chamber.

*4.7c—Visual Avoidance:* Visual avoidance behavior was assessed using an assay modified from [15,33]. Tadpoles were screened on Day 0 for the optomotor response to evaluate ability to respond to visual stimuli. The optomotor response test is administered by transferring 6 tadpoles at a time to a clear-bottomed plexiglass tank, placed on a clear plexiglass platform which sits over a projector (3M). Animals are placed in the center, then given 1 min to distribute throughout the tank. Visual stimuli in the form of alternating black and white bars that move unilaterally were projected onto the bottom of the arena. Animals that respond to stimuli by moving in the direction of the moving bars, as described by [70], were used in the visual avoidance experiments; animals who did not respond were eliminated from further behavioral testing. For the visual avoidance assay, tadpoles were placed individually in a clear, rectangular plexiglass arena (20 × 8 cm), atop a clear plexiglass platform that sits over a projector. The entire setup was enclosed by a large box to prevent outside light from entering the arena. Each animal was placed into the center of the arena and given 1 min to move throughout the arena; visual stimuli were then projected onto the bottom of the arena. Randomly distributed white dots (0.8 cm in size) were projected on a dark grey background, and the dots moved from left to right in the arena. Visual stimuli were generated using Microsoft PowerPoint’s animation function. Tadpole movement and visual stimuli were recorded using EthoVision software. As in [15], an interaction was defined as perpendicular contact between a moving spot and the tadpole’s field of vision. The first 10 interactions with moving spots were counted, and avoidance was scored when the tadpole displayed a sharp turn within less than one second of contact with the stimulus.
$$Avoidance\;Index=\frac{scored\;avoid\;maneuvers}{first\;10\;dot\;interactions}$$

Behavior was recorded for 2 min, beginning at Day 0 (which represents a pre-treatment baseline) and repeating the procedure over the course of 7 days, excluding one recovery day immediately following injury. This value was then plotted as an avoidance index, and Days 2–8 were normalized to the avoidance index baseline collected on Day 0.

### 4.8. Statistics

Statistical analysis was performed using R Studio (version 1.4.1106). Where applicable, error bars express the standard error of means for experiments. Two-way ANOVA followed by Tukey’s post hoc was used to determine whether sham and injured brains were significantly different in their percent brain water content (edema). The same statistical test was applied to all behavior data (Open Field C-Starts, Light/Dark Box, and Visual Avoidance assays). For the Visual Avoidance assay, data are presented as a mean +/− SEM. For all behavioral assays, *n* values are based on a power analysis performed using R Studio. The number of animals used (*n*) is indicated in each figure legend. Levels of significance are indicated using the following notation: * *p <* 0.05; ** *p <* 0.01; *** *p <* 0.001; NS = no significant difference. For all experiments, a *p* value of < 0.05 was considered to indicate a significant difference.

## Figures and Tables

**Figure 1 ijms-23-07578-f001:**
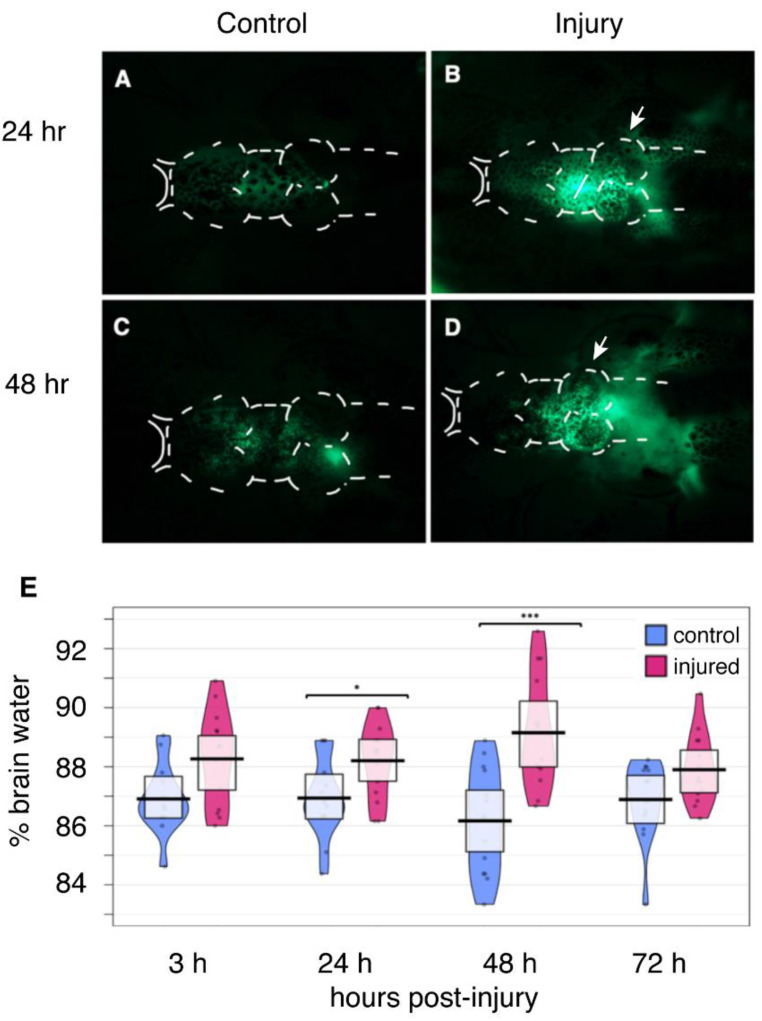
**Focal Impact Injury leads to Edema and Disruption of BBB integrity**. Sham tadpoles were anesthetized and transferred temporarily to the injury cradle prior to intraventricular injection with sodium fluorescein (NaF) either 24 h post-injury (h.p.i.) (**A**) or 48 h.p.i (**C**). Injured animals were anesthetized and then subjected to focal impact injury at the optic tectum (white arrows). Following either 24 h (**B**) or 48 h (**D**) of recovery post-injury, tadpoles were given an intraventricular injection with 10 nL of 1 μg/mL NaF; diffusion of the tracer was visualized using fluorescence microscopy. Arrows in (**B**,**D**) show the site of injury. (**A**–**C**) White dotted lines indicate the outer borders of the brain. Images are representative of *n* = 6 animals per treatment group for each time point. (**E**) * *p* < 0.05; *** *p* < 0.001.

**Figure 2 ijms-23-07578-f002:**
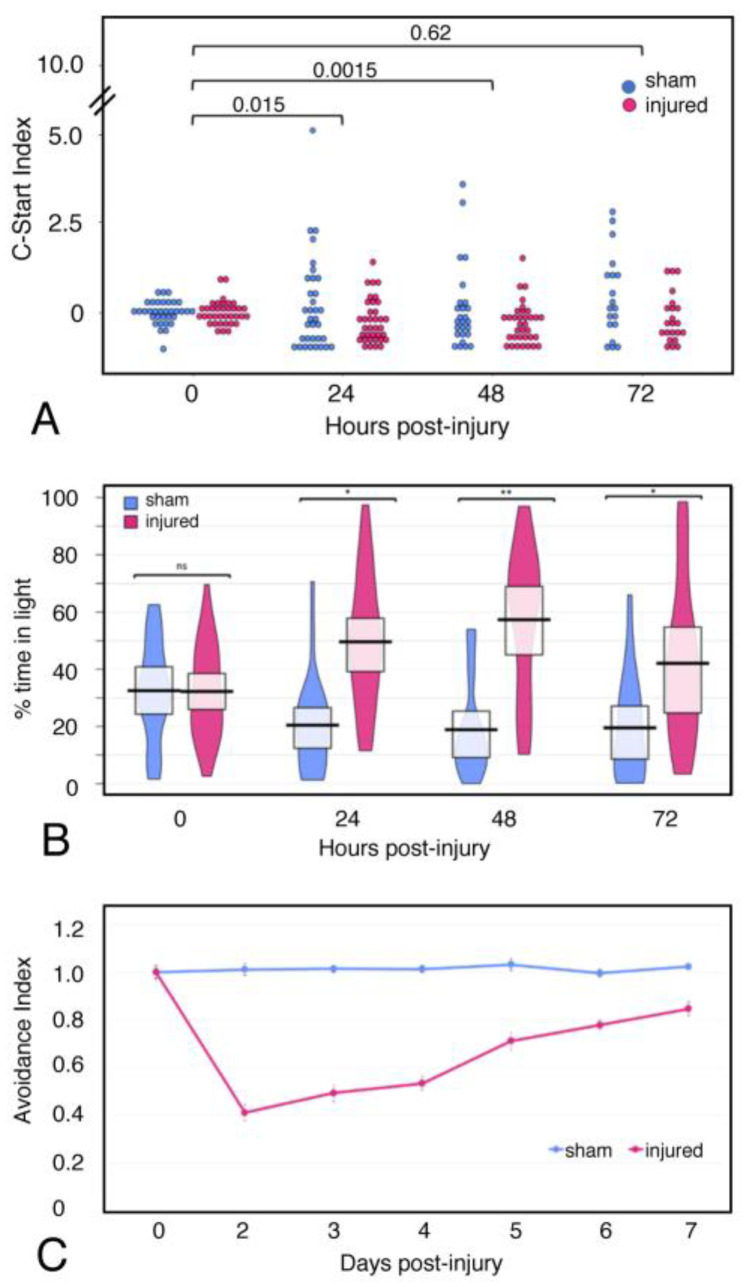
**Focal impact injury causes deficits in visually mediated behaviors.** NF stage 55 *Xenopus laevis* tadpoles were evaluated for baseline behavior at 0 h.p.i. in each behavioral assay, then a focal impact injury was administered to the optic tectum (OT) of each animal in the injured group (pink). Sham animals (blue) were anesthetized and moved in and out of the injury cradle after Day 0. (**A**) **Open field testing, C-Start reflex.** Open field tests were performed once per day up to 72 h.p.i. following a 0 hpi baseline. The number of C-Start reflexes performed in the injured group was significantly reduced at 24 and 48 h.p.i. (**B**) **Light/Dark box assay.** Open field tests were performed once per day up to 72 h.p.i. following a 0 hpi baseline. Light zone time was significantly increased in injured animals at all three timepoints. (**C**) **Visual avoidance behavioral assay**. Baseline data were collected at 0 h.p.i., then the injury or sham injury was administered. The test was performed daily to assess the recovery of the visual avoidance behavior. Beginning 48 h.p.i., injured tadpoles display a significant deficit in the avoidance behavior, which they appear to recover over a time course of 7 d.p.i. [Significant *p*-values from two-way analysis of variance (ANOVA) are shown as * *p*  <  0.05, ** *p*  <  0.01, n.s. indicates no significant difference].

**Figure 3 ijms-23-07578-f003:**
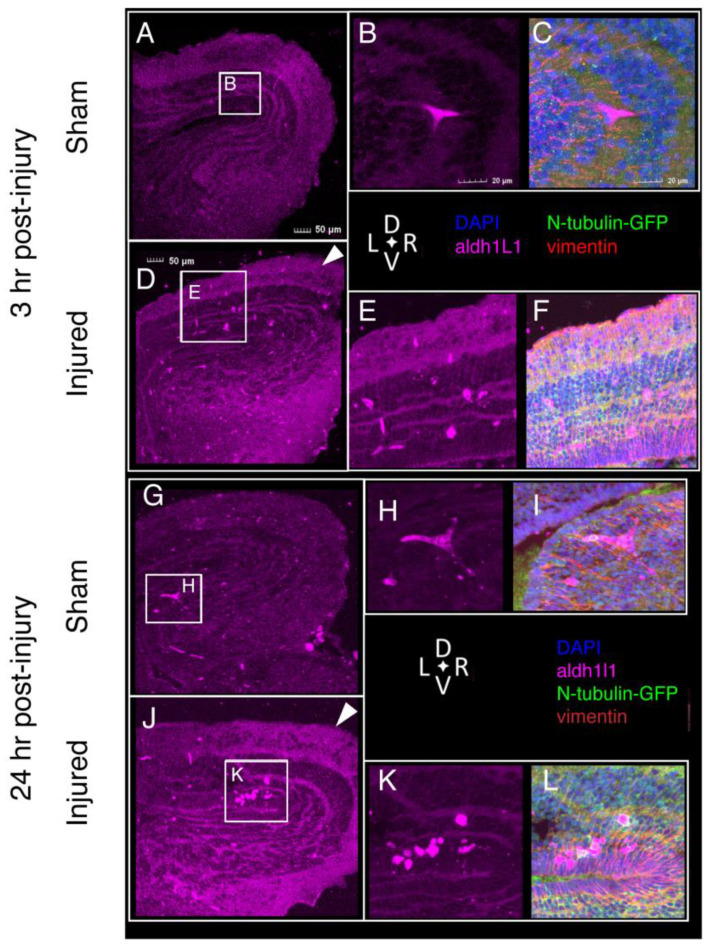
**Astrocytes visualized with antibodies against Aldh1L1 via immunofluorescence microscopy.** Sections from NF stage 55 N-tubulin-GFP^+^ *Xenopus laevis* midbrains of both sham (**A**–**C**) and injured (**D**–**F**) animals at 3 h post-injury show astrocytes (*aldh1L1*-positive) in the dorsolateral midbrain and optic tectum (OT). Astrocytes in midbrains 3 h after injury show differences in morphology (**B** vs. **E**) and distribution (**A** vs. **D**); these differences persist at 24 h post-injury. (**G**–**I**), sham; (**J**–**L**), injured. (For all images, Alexa Fluor 647 signal was digitally converted to magenta. Sections were also processed to visualize neurons (anti-GFP, green), nuclei (DAPI, blue), and intermediate filaments (anti-Vimentin, red). Sections are 14 µm thick and images selected are representative of all samples examined (*n* = 6 brains).

**Figure 4 ijms-23-07578-f004:**
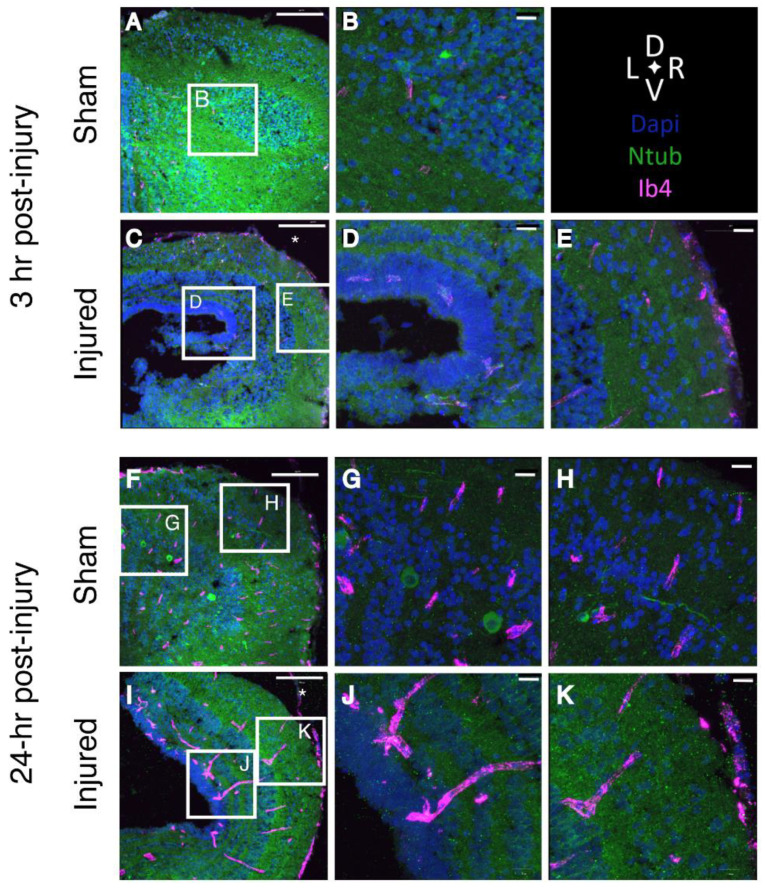
**Microglia localize around the ventricle of the injured midbrain.** Confocal images of midbrains 3 h after injury; midbrains were isolated from N-tubulin-GFP tadpoles (neurons, green) injected with Ib4-Alexa 647 to label microglia (magenta). Brain were sectioned at 15 µm; nuclei are labeled with DAPI (blue). (**A**,**B**) Sham-treated midbrain. (**C**–**E**) Midbrain 3 h after injury. (**E**) Microglia accumulate near the site of injury. (**A**,**C**): Scale bar 100 µm. (**B**,**D**,**E**): Scale bar 20 µm. Scale 100 µm. (**F**–**K**), confocal images of St. 55 midbrains after 24 h. In the sham brain, microglia are visible both close to the ventricle (**G**) and toward the periphery of the neuropil (**H**). In the injured midbrain, microglia form clusters both near the ventricles (**J**) and in the neuropil near the site of injury (**K**). (**F**,**I**): Scale bar 100 µm. (**G**,**H**,**J**,**K**): Scale bar 20 µm. Asterisks in (**C**,**I**) mark the site of injury (*n* = 3). * indicates the site of administration for the focal impact injury.

**Figure 5 ijms-23-07578-f005:**
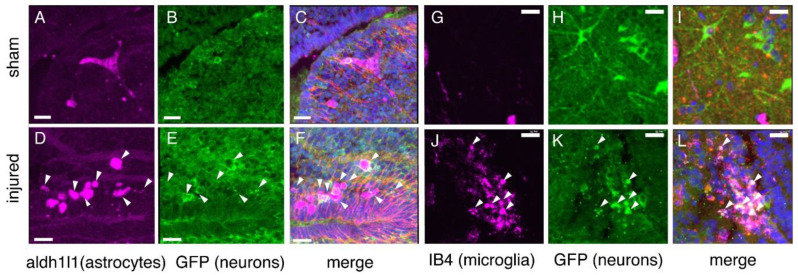
**Evidence of phagocytic astrocytes and microglia following injury**. Injured midbrains 24 h post-injury include both astrocytes (**D**–**F**) and microglia (**J**–**L**) that are positive for N-tubulin-GFP (white arrowheads), as well as for aldh1l1 (astrocytes) or IB4 (microglia); merged fluorescent signal appears white. Magenta (**A**–**F**), aldh1l1; Magenta 9 (**G**–**L**), IB4; Green, N-Tubulin-GFP; Blue, DAPI. Sham brains (**A**–**C**,**G**–**I**) contain no more than a few such cells. scale bars, 20 µm.

**Figure 6 ijms-23-07578-f006:**
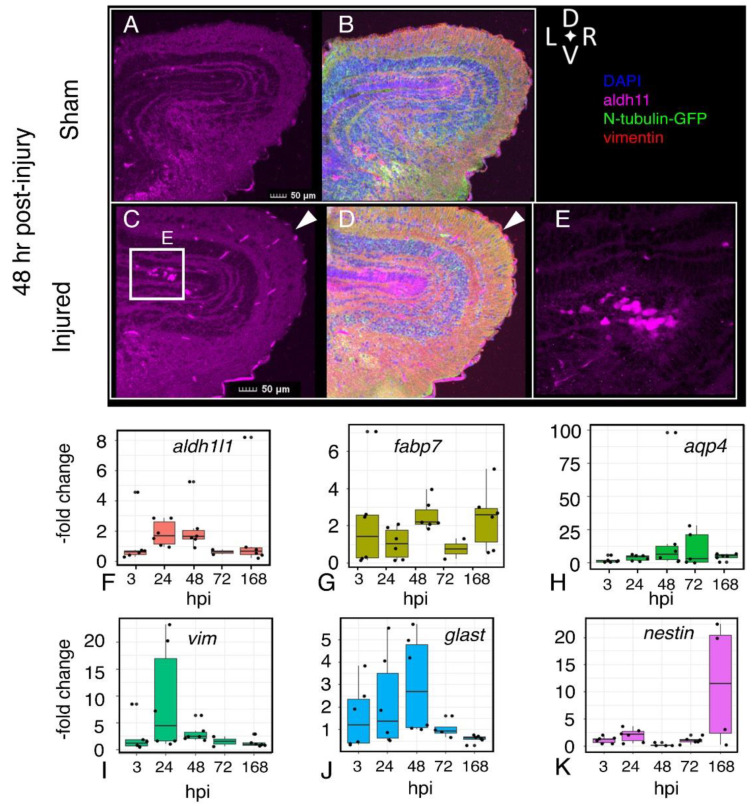
**Persistence of altered astrocyte morphology and expression of astroglial genes following injury**. Visualization of astrocytes (aldh1l1-positive cells, magenta), neurons (N-tubulin-GFP), radial glia (vimentin-positive, red) in sham (**A**,**B**) and injured (**C**–**E**) 48 h after injury. Amoeboid astrocytes accumulate near the ventricular layer (**C**,**E**). Quantitative RT-PCR assays for expression of astroglia-associated genes *Aldehyde dehydrogenase 1 family member 1 (aldh1l1)* (**F**), *fatty acid-binding protein 7 (fabp7)* (**G**), *aquaporin4 (aqp4)* (**H**), *vimentin (vim)* (**I**), *excitatory amino acid transporter 1 (glast)* (**J**), and *nestin (nes)* (**K**). RNA was isolated from sham and injured midbrains at the intervals shown; injured values are normalized to sham-treated controls. Plots show individual data points and a box showing the mean and 95% confidence intervals. *n* ≥ 6 experiments.

**Figure 7 ijms-23-07578-f007:**
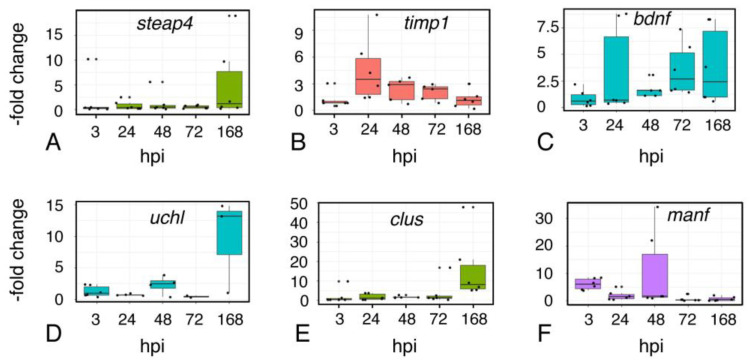
**Expression of genes associated with astrocyte reactivity and neuroprotection.** Quantitative RT-PCR assays for expression of the “reactive astrocyte”-associated genes steap4 (**A**), and *timp1* (**B**), as well as the neuroprotective genes *brain-derived neurotrophic factor (bdnf)* (**C**), *ubiquitin C-terminal hydrolase-1 (uchl)* (**D**), *clusterin (clus)* (**E**), and *mesencephalic astrocyte-derived neurotrophic factor (manf)* (**F**). RNA was isolated from sham and injured midbrains at the intervals shown; injured values are normalized to sham-treated controls. Plots show individual data points and a box showing the mean and 95% confidence intervals. *n* ≥ 6 experiments.

**Figure 8 ijms-23-07578-f008:**
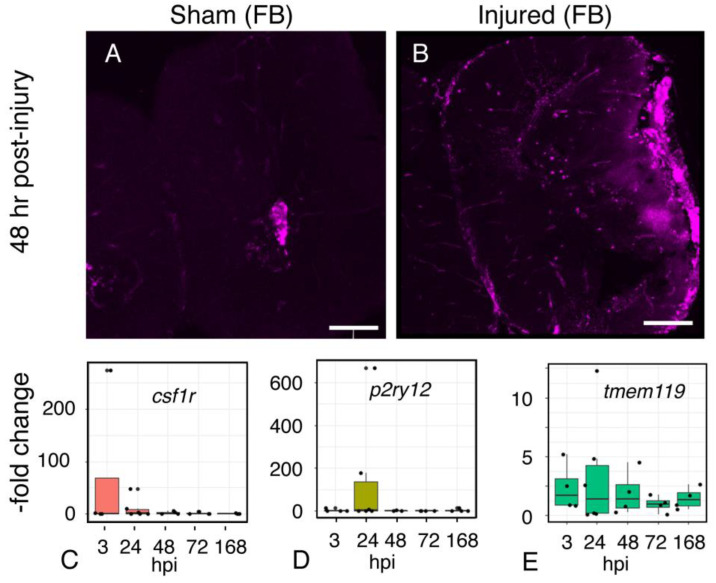
**The microglial response to midbrain injury extends to the forebrain.** Confocal images of microglia (IB4-positive cells) in sham (**A**) and injured (**B**) forebrains 48 h after injury. Microglia accumulate within the ipsilateral forebrain, at a distance from the injury site at the optic tectum (not pictured). The images are characteristic representations of all samples (*n* = 3). Brain sections are 50 µm thick. Scale 100 µm. (**C**–**E**), Time course of expression of microglia-associated genes following focal impact injury. RNA was isolated from sham and injured midbrains at the intervals shown; injured values are normalized to sham-treated controls. Plots show individual data points and a box showing the mean and 95% confidence intervals. N ≥ 4 experiments.

**Figure 9 ijms-23-07578-f009:**
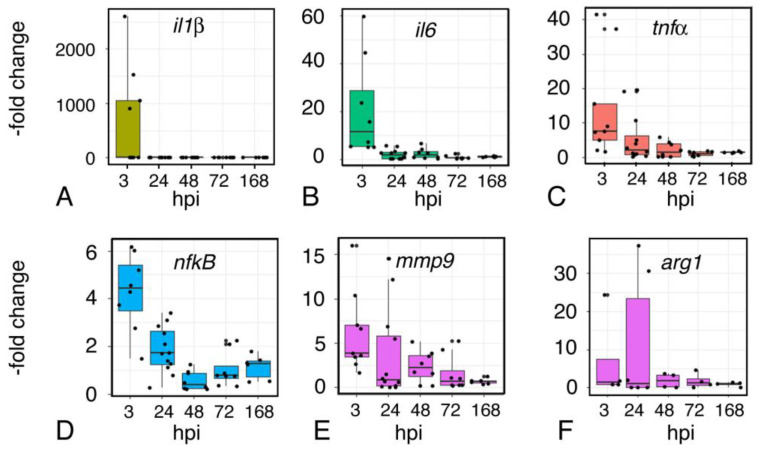
Time course of expression of inflammation-associated genes following injury. Quantitative RT-PCR assays for expression of the inflammatory cytokines Interleukin1-β (il1β) (**A**), Interleukin6 (il6) (**B**), Tumor Necrosis Factor-α (tnf-α) (**C**), as well as Nuclear Factor kB (nfkB) (**D**), matrix metalloproteinase 9 (mmp9) (**E**), and arginase (arg) (**F**). RNA was isolated from sham and injured midbrains at the intervals shown; injured values are normalized to sham-treated controls. Plots show individual data points and a box showing the mean and 95% confidence intervals. N ≥ 4 experiments.

**Figure 10 ijms-23-07578-f010:**
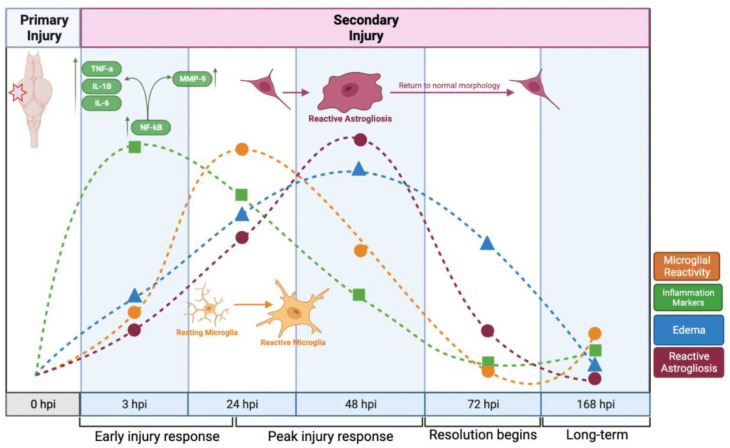
**A timeline of response to focal impact injury**. The response to focal impact injury begins with the release of inflammatory cytokines, followed by microglial activity, astrocyte reactivity, and edema. Within 48 h, each of these activities has peaked, and all have subsided to near-baseline levels after one week (This image was made using BioRender (Suite 200, Toronto, ON, M5V 2J1)).

**Table 1 ijms-23-07578-t001:** Survival of Injured and Sham-treated Tadpoles following Focal Impact Injury.

Time (h.p.i)	Sham Death	Injured Death	Sham Total	Injured Total	% Survival Sham	% Survival Injured
0	0	0	191	206	100	100
3	1	0	190	206	>99%	100
24	1	2	170	184	>99%	98%
48	0	1	105	108	100%	99%
72	0	1	73	81	100%	98%
168	0	2	30	30	100%	93%

## Data Availability

Not applicable.

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
