# Peer review of "A Focal Impact Model of Traumatic Brain Injury in Xenopus Tadpoles Reveals Behavioral Alterations, Neuroinflammation, and an Astroglial Response"

_ijms, 2022, doi:10.3390/ijms23147578_

Round 1

Reviewer 1 Report

In general, I support publication of this manuscript. However, extensive changes must be undergone as outlined below.

It was difficult to go through the manuscript and separate the sequence of events, as the authors have presented an unorderly manuscript. This must be corrected by any means.

Preliminary experiments. These should be developed in one subsection only. As it is now, it not clear where the preliminary findings end and where the main results start, thus creating confusion. The subsection must clearly labelled to indicate that preliminary studies are described. No findings must be present in the main text. All the details must be presented in supplementary material. More findings than just figures should be presented. Authors will need to provide to readers all details to support their main studies.

M & M

Please provide a separate section with all the data management and the analysis performed. As these are now scattered, it is difficult to evaluate with a global view.

After making the above improvements, please resubmit for further evaluation.

Author Response

IJMS- 1732262

Spruiell Eldridge, Teetsel, Torres et al., “A focal impact model of traumatic brain injury in Xenopus tadpoles reveals behavioral alterations, neuroinflammation, and an astroglial response”

Response to Reviewer 1:

We thank the reviewer for their comments and suggestions.  In response to their reviews, we have made the following revisions:

R1:  It was difficult to go through the manuscript and separate the sequence of events, as the authors have presented an unorderly manuscript. This must be corrected by any means.

  • we have extensively reorganized section 2.7 (pp 10-11), “Tissue damage and cellular response to injury”, so that it is easier to follow;
  • we have modified Fig. 5 to include both single-channel and merged images at one time point (p 14);
  • we have modified Fig. 6, adding aquaporin4 to replace aldolase C in our Q-RT-PCR panel (p 15);
  • we have added a summary figure (Fig. 10, p 21) illustrating the time course of response to injury;
  • we have expanded the discussion to address the relationship between different components of this response (p 20, lines 546-580);
  • We have revised the text throughout to improve clarity.

R1: Preliminary experiments. These should be developed in one subsection only. As it is now, it not clear where the preliminary findings end and where the main results start, thus creating confusion. The subsection must clearly labelled to indicate that preliminary studies are described. No findings must be present in the main text. All the details must be presented in supplementary material. More findings than just figures should be presented. Authors will need to provide to readers all details to support their main studies.

  • we have revised and expanded sections 2.1 and 2.2 (pp 4-5) to provide a clear distinction between our initial studies on younger stage, as well as the rationale for conducting subsequent experiments on the older (st. 55) tadpole;

R1:M & M - Please provide a separate section with all the data management and the analysis performed. As these are now scattered, it is difficult to evaluate with a global view.

  • we have expanded our discussion regarding statistical methods (p 25, section 4.8);

We hope that these additions and modifications will improve the clarity of the manuscript.

Reviewer 2 Report

Indeed, traumatic brain injury (TBI) is a global cause of disability and we lack effective therapies that help restore and restore the nervous system. The authors studied the response of Xenopus laevis tadpoles to focal impact damage to the optic nerve cover as a model of TBI. The authors' results demonstrate that the Xenopus response to focal impact injury resembles cellular changes observed in rodents, the Xenopus tadpole offers a novel and scalable vertebrate model for mammalian TBI. I really liked the work of the authors, the experiment was intelligently planned, the results are logically and consistently presented, the illustrative material was well chosen. I think that the article deserves to be published in its current form.

Author Response

We thank Reviewer2 for their kind and appreciative comments.  We have sought to improve the manuscript through the following revisions:

  • we have revised and expanded sections 2.1 and 2.2 (pp 4-5) to provide a clear distinction between our initial studies on younger stage, as well as the rationale for conducting subsequent experiments on the older (st. 55) tadpole;
  • we have extensively reorganized section 2.7 (pp 10-11), “Tissue damage and cellular response to injury”, so that it is easier to follow:
  • we have modified Fig. 5 to include both single-channel and merged images at one time point (p 14);
  • we have modified Fig. 6, adding aquaporin4 to replace aldolase C in our Q-RT-PCR panel (p 15);
  • we have expanded our discussion regarding statistical methods (p 25, section 4.8);
  • we have added a summary figure (Fig. 10, p 21) illustrating the time course of response to injury;
  • we have expanded the discussion to address the relationship between different components of this response (p 20, lines 546-580).

We hope that these additions and modifications will improve the clarity of the manuscript.

Round 2

Reviewer 1 Report

The manuscript has been improved.
Before publication, the authors should revise the discussion by going into greater depth into the issues developed therein, as well as by adding some recent relevant references which will help to interpret better the current findings.

Author Response

We thank Reviewer 1 for their additional comments.  We have

  • expanded the Discussion to incorporate some thoughts on the relationship between edema and elevated expression of aquaporin4, in the context of published findings about cytotoxic and vasogenic edema.
  •  introduced additional references of more recently published work.  
  • As the Discussion section was already fairly long, we have also streamlined this section to accommodate these additions.